# Double Gain: The Radio Frequency Catheter Ablation of Ventricular Aneurysm Related Recurrent Ventricular Tachycardia on a Tremendous Cardiac Outpouching

**DOI:** 10.3390/diagnostics12081955

**Published:** 2022-08-12

**Authors:** Kexin Li, Yufeng Jiang, Ziyin Huang, Yafeng Zhou

**Affiliations:** 1Department of Cardiology, The First Affiliated Hospital of Soochow University, Suzhou 215006, China; 2Department of Cardiology, Dushu Lake Hospital Affiliated to Soochow University, Suzhou 215000, China; 3Institution for Hypertension of Soochow University, Suzhou 215000, China

**Keywords:** idiopathic dilated cardiomyopathy, ventricular aneurysm, radio frequency catheter ablation, sustained polymorphic ventricular tachycardia

## Abstract

Dilated cardiomyopathy (DCM) is a classic type of non-ischemic cardiomyopathy. Of these, idiopathic cardiomyopathy (IDCM) is a rare type of non-genetic dilated cardiomyopathy. More specifically, the patient had suspected IDCM combined with sustained polymorphic ventricular tachycardia (PMVT) of left ventricular basal segmental origin, cardiac systolic dysfunction and an ejection fraction (EF) of 29%. He had an abnormally large ventricular aneurysm (VA) in the posterior wall of the left ventricle with left ventricular end diastolic dimension (LVDd) of 90 mm. We performed an endocardial radiofrequency catheter ablation (RFCA) of the patient’s recurrent ventricular tachycardia (VT) on the basis of an implantable cardioverter (ICD). Although minimally invasive RFCA also carries a high risk, it is currently a two-pronged option to improve the patient’s quality of life and to prevent the recurrence of VT. Postoperatively, the patient was routinely given optimal anti-arrhythmic and heart failure (HF) treatments to improve cardiac function as well as being followed up for 9 months. The patient’s EF ascended to 36% without any recurrence of VT. In summary, RFCA of suspected IDCM combined with VA and VT of basal area origin would be an effective treatment.

## 1. Introduction

DCM is a primordial myocardial disease of uncertain cause. It is distinguished by enlargement of the left or right ventricle or bilateral ventricles with ventricular systolic dysfunction, with or without congestive heart failure. Ventricular or atrial rhythm irregularities are predominant, with progressive deterioration, and mortality can occur at any stage of the disease [1]. Left VAs, particularly idiopathic VAs, are rare in adults and cause discrete thinning of the ventricular wall, resulting in localized ventricular outpouching [2]. For PMVT due to DCM, RFCA is an alternative option where drugs and external electrical resuscitation are not effective [3]. The novelty of this case was that the left ventricle was huge like a balloon with VA, and there was no recurrence of VT at 9 months of continuous follow-up after endocardial RFCA of VT was performed.

## 2. Case Reports

A 67-year-old male disclaimed hypertension and diabetes mellitus, had quit smoking for more than 1 year, did not abuse alcohol and denied a family history of cardiac disease, such as DCM and coronary heart disease (CHD). The patient went to another hospital in 2020 with the complaint of “recurrent chest tightness for more than 1 year and palpitations for 1 day”. The three-dimensional echocardiography (3D-echo) showed an enlarged heart and a reduced EF. The coronary angiogram showed a few plaque infiltrations in the middle and distal segments of the RCA with a gross wall. An ICD (St. Jude CD1231-40) was subsequently implanted for prevention of sudden cardiac death (SCD), and the patient was discharged from the hospital with optimization of drug therapy for heart failure (HF). In the fifth month after ICD implantation, the patient experienced recurrences of chest tightness and palpitations and was subsequently admitted to our hospital. Admission electrocardiogram (ECG) showed sinus and ventricular pacing rhythm (Figure 1a), and the 3D-echo showed basal VA formation in the posterior left ventricular wall, left ventricular diameter (LVD) of 90 mm, and LV ejection fraction (EF) of 29% (Figure 2a). Neither laboratory tests nor cardiac angiography (CAG) showed any apparent abnormalities, and exceptional serum N-terminal pro-brain natriuretic peptide (NT-proBNP) level was 1705 pg/mL.

On admission, the patient still experienced recurrences of chest tightness and palpitations, and confirmed the recording of the ICD programmer, which showed that there was a series of VT events, and the first event was recorded as a frequency of 218 beats per min (bpm) for a duration of 29 s. There were 15 repeated ICD discharges with different degrees of distress before the RFCA procedure.

The patient was given anti-arrhythmic therapy with Amiodarone and Mexiletine on the basis of adequate optimization of HF medication (ARNI, β-blockers, MRA, SGLT2i were titrated to maximum tolerated dose). After the treatment with extracorporeal synchronous direct current resuscitation, the VT was still not well controlled. Therefore, the anti-arrhythmic drugs were adjusted to administer Amiodarone 0.2 g qd and Sotalol 80 mg qd orally, and intravenous Amiodarone, Esmolol and Lidocaine were given intermittently to counteract the VT, but there were still recurrent VT episodes (Figure 1b). The patient had several episodes of VT with a heart rate of 130–150 bpm during his hospitalization, which did not reach the ICD autoregulation threshold. The VT caused significant chest tightness, shortness of breath and a critical hemodynamic state, and the patient was treated with electrical cardioversion. Since surgical resection for VA was a risky procedure due to the high location of VA, and the heart was huge like a balloon, RFCA was now the optimal approach to control VT and improve prognosis. After full communication with the patient and his family, our treatment team decided to treat him with RFCA.

The procedure of RFCA lasted for 5 h. The right femoral vein and femoral artery were punctured and the large ST-SF ablation head and right ventricular quadrupole electrode were delivered. In sinus rhythm (SR), the left ventricular late potential zone was marked, and a large area of low voltage was seen in the inferior, posterior basal and middle walls of the left ventricle for local late potential linear ablation (Figure 3a); VT3 was again induced with ventricular stimulation, continuing the previous ablation line and late potential ablation (Figure 3b). Hence, ventricular tachycardia of the patient was confirmed as PMVT. Three-dimensional reconstruction of the coronary CT angiography (CCTA) showed the shape of the VA (Figure 3c). Ventricular stimulation with S1S1 500 ms/280 ms failed to induce VT, and the procedure was completed.

Subsequent ECG is shown in (Figure 1c). The cardiac enzyme profile was rechecked 5 days after the operation: high sensitivity troponin T (hs-cTnT (723 pg/mL)) and NT-proBNP (541 pg/mL). A 3D-echo prior to discharge still showed posterior left VA formation (Figure 2b). The patient was discharged on the 12th postoperative day with medication to optimize anti-arrhythmic and HF drugs.

At consecutive monthly outpatient follow-up visits, the most recent on 18 May 2022, the patient had no recurrence of VT or ICD discharge (Figure 1d), and the 3D-echo described EF of 36% and LVDd of 80 mm (Figure 2c) 9 months after the operation on 26 August 2021. The patient’s EF had increased from 29% to 36% in 9 months, showing a significant improvement in cardiac function. The present study was approved by the Ethics Committee of Soochow Dushu Lake Hospital.

## 3. Discussion

The formation of VA is usually associated with myocardial scarring after myocardial infarction, but a minority of non-ischemic VAs may be associated with genetics, infection, sarcoidosis and other causes. In general, ischemic VAs tend to occur in the apical region and the anterior wall, while non-ischemic VAs tend to be found in the posterior wall and basal segments [4]. In addition, VT is a serious life-threatening manifestation. It is generally agreed that DCM, especially in its advanced stage, may present with a variety of arrhythmias, either supraventricular or ventricular, or with atrioventricular block (AVB), complete left bundle branch block (LBBB), or even atrial fibrillation (AF). Of all the types of arrhythmias, the most prevalent is tachyarrhythmia, which can lead to SCD. The causes of arrhythmias are, initially, an enlargement of the heart and a decline in cardiac function. It is due to the activation of the neurohormonal state, especially the RAAS and the SAS system. In addition, especially in patients with advanced HF, there is often a conjunction of hypokalemia and hypomagnesemia, which is more likely to trigger arrhythmias. At the same time, the use of drugs, such as cardiac stimulants or diuretics, can also induce arrhythmias [5]. However, the patient’s VT was recurrent despite the use of optimal oral drugs for control of VT. Intermittent intravenous Esmolol was also given to further control sympathetic hyperexcitability. The mechanism for the intermittent intravenous administration of Esmolol in addition to oral Sotalol is that β-blockers reduce sympathetic excitability and play an important role in reducing the recurrence of VT or ventricular fibrillation (VF) [6]. Even if the patient is already on β-blocker therapy, an episode of VT or VF requires the addition of an intravenous β-blocker to the original β-blocker in order to block sympathetic hyperexcitability. Therefore, for a patient with recurrent VT, treatment on the basis of oral Sotalol and intravenous esmolol can block sympathetic excitation and reduce the recurrence of VT. Unfortunately, the patient still continued to have recurrent episodes of VT after the use of intravenous Esmolol. This forced us to choose the optimal RFCA procedure to alleviate the anguish of the patient.

Moreover, sustained polymorphic VT is known as PMVT when the duration of VT is ≥30 s, or when the duration is <30 s but the VT is accompanied by hemodynamic disturbances requiring early intervention. PMVT in combination with structural heart disease (SHD) is most commonly seen in coronary artery disease (CAD), followed by DCM, arrhythmogenic right ventricular cardiomyopathy (ARVC), hypertrophic cardiomyopathy, complex congenital heart disease, valvular disease and myocarditis. Patients with hereditary arrhythmia syndromes such as long QT syndrome (LQTS), short QT syndrome (SQTS), Brugada syndrome and early repolarization syndrome (ERS) do not have structural changes in the heart but often experience PMVT, mostly associated with genetic abnormalities [5].

The expert consensus in 2019 states that catheter ablation at the trigger site of PMVT is recommended as Class II a, Level of Evidence B, for patients with SHD. However, most of these patients require the epicardial route, which is a relatively complex and risky procedure, and the results of long-term ablation success remain poorly studied. Of note, RFCA of VT in patients with SHD has a <5% incidence of procedural complications, mainly AVB, cardiac perforation, stroke or transient ischemic attack (TIA), HF or even fatality [3].

Patients with a family history of SCD, conduction system disease or skeletal muscle disease are often prone to DCM, which is associated with mutations in the cytoskeleton, myosin/Z-band, nuclear membrane and intercalator protein genes, most of which are autosomal dominant, although autosomal recessive and X-linked inheritance are also prone to DCM [7]. When DCM is combined with mutations in LMNA, EMD, SCN5A, etc., conduction system disease and supraventricular arrhythmias are susceptible. Of these, the presence of supraventricular arrhythmias (premature atrial contraction, atrial tachycardia, atrial flutter (AFL) and atrial fibrillation (AF)), if observed in patients with dilated cardiomyopathy, should prompt investigation of familial LMNA cardiomyopathy, which is present in 73% of carriers of these genes, compared to 36% of patients without this genetic defect [8]. In addition, about 10% of patients with dilated cardiomyopathy present with AF [9,10]. Certain mitochondrial diseases and inherited metabolic disorders, hypothyroidism, chemotherapeutic drugs such as anthracyclines and tyrosine kinase inhibitors, and non-familial factors such as nutritional deficiencies have been associated with the development of DCM. Atrioventricular node folding and bypass-related tachycardia are not usually associated with DCM, while sustained atrial tachycardia may be a cause rather than a result of left ventricular dysfunction [11]. Clinically, DCM can present with a variety of arrhythmias, even malignant arrhythmias leading to SCD; on the other hand, sustained tachycardia and frequent premature ventricular contraction can lead to cardiomyopathy or exacerbate pre-existing cardiac dysfunction.

A growing number of biometric analyses and high-throughput platform studies suggest that DCM may have one or more possible genetic mutations, such as TNN, which is more clearly associated with familial DCM, and LMNA, which predisposes to AVB and also to Limb-Girdle myopathy [1,12]. Circulating Ca^2+^ protein imbalance leads to the development of IDCM, the incidence of which depends on the species [13,14]. The main molecular mechanisms in the pathogenesis of DCM are increased circulating and tissue levels of norepinephrine, aldosterone, endothelin, vasopressin and cytokines, which lead to structural changes in cardiomyocytes and the extracellular matrix, ultimately leading to myocardial cell necrosis and fibrosis [15]. These microscopic changes lead to adverse structural remodeling, which is prone to malignant events such as SCD, VT and VF due to disturbances in the conduction system of the remodeled cardiac structure [16], so early diagnosis and intervention is very essential to save lives. The diagnosis of DCM used to be based on myocardial biopsy and imaging CMR; nowadays, there is a trend towards molecular biology and etiology diagnosis and further precise treatment such as immune-targeted therapy and cell therapy [7]. This case was a non-ischemic cardiomyopathy, and it was presumed that if genetic testing was performed, one of the main immunomarkers for anti-myocardial antibodies (AHA) might be positive. Testing for AHA in this patient with non-ischemic DCM allowed for targeted selection of immunotherapy and prediction of risk of SCD and mortality in DCM based on the test findings. Positive AHA were detected in 42–85% of idiopathic DCM. Positive L-type calcium channel antibodies and anti-adrenergic β1 receptor antibodies are independently predictive of death in heart failure and SCD in DCM. Immunosorbent therapy (IgA/IgG) can be used in patients positive for AHA [5]. However, many patients do not have the conditions to receive molecular diagnosis and precise treatment. On the one hand, there is an imbalance in medical resources, and on the other hand, patients with sudden progression of the disorder are likely to suffer from SCD. In this context, the preventive treatment of DCM is more often based on pharmacological treatment such as MRA, β-blockers, ARNI, ACEI to improve or reverse ventricular remodeling, early application of anti-arrhythmic drugs and ICD in high-risk patients before the EF falls below 35% to prevent SCD events [16]. Moreover, it is an established fact that the gold standard for the pathological diagnosis of cardiomyopathy is the myocardial biopsy. However, obtaining a piece of tissue off a sizable and skinny piece of myocardium would be extremely hazardous. Although a myocardial biopsy could not be performed for pathological typing and diagnosis in this case, a clinical exclusionary diagnosis was made, and IDCM considered. Although this patient had a poor myocardial condition and was at risk of cardiac rupture, RFCA might reduce the likelihood of malignant adverse events (e.g., SCD) as well as further improve the patient’s quality of life and prognosis.

Additional multimodal diagnostic models for early diagnosis have been investigated and await further validation [17]. However, these treatments can only slow the progression of the disease, and the ultimate treatment remains a heart transplant (HT). The scarcity of HT supplies, the difficulty of the procedure and post-operative rejection are also factors that affect the five-year survival rate [18]. If a patient has VT in combination with DCM and an EF of less than 35%, RFCA can be a two-pronged treatment for the patient under such demanding and difficult conditions, although many studies suggest that the five-year survival rate and the recurrence rate of VT are not statistically significant in relation to HT [3].

In the case presented, the patient was suffering from suspected IDCM combined with a VA and frequent episodes of VT of VA origin, the heart was extremely huge with hemodynamic instability, and repetitive ICD discharges failed to restore SR. In addition, the patient’s VA was close to the base of the left ventricle near the aortic root, the surgical excision of the VA was extremely hazardous, and surgical excision itself would cause arrhythmias [19]. At this time, only RFCA in such an extreme condition could have been a boon to the patient rather than a constant fear of onset of SCD after VT for the patient and his relatives. The procedure seemed to be a glimmer of light to save the patient from the predicament [20]. However, there are risks attached to everything, and it was very difficult to hold on to this light because of the long ablation time, the large ablation area and the risk of heart failure during the procedure. Fortunately, the ablation was successful and the patient was followed up 9 months after the procedure with no further episodes of VT such as being unsuitable for ICD discharge.

This patient did not experience recurrence after intraventricular ablation of VT, and the results of a multicenter study with ARVC treatment suggest that combined endocardial and epicardial ablation showed better VT-free survival than endocardial ablation alone [21]. However, our case was of VT of left VA origin in a large heart with endocardial ablation and no recurrence of VT and ICD discharge events at 9 months post-procedure follow-up. This case was a paradox compared to most research findings and had a successful outcome. Follow-up will continue, and genetic testing will be recommended for the patient, as well as his first- and or second-degree relatives. Cases will continue to be collected, as successful endocardial ablation and the absence of recurrent VT for 9 months has useful implications for learning. However, this was only one case report without clear medical evidence. Future studies of endocardial ablation of IDCM combined with VT could be conducted in a joint multicenter setting.

## 4. Conclusions

Overall, non-recurrence of VT within 9 months of intraventricular RFCA of PMVT in a large heart of unknown etiology with suspected IDCM is extremely rare and indicative of successful clinical management.

## Figures and Tables

**Figure 1 diagnostics-12-01955-f001:**
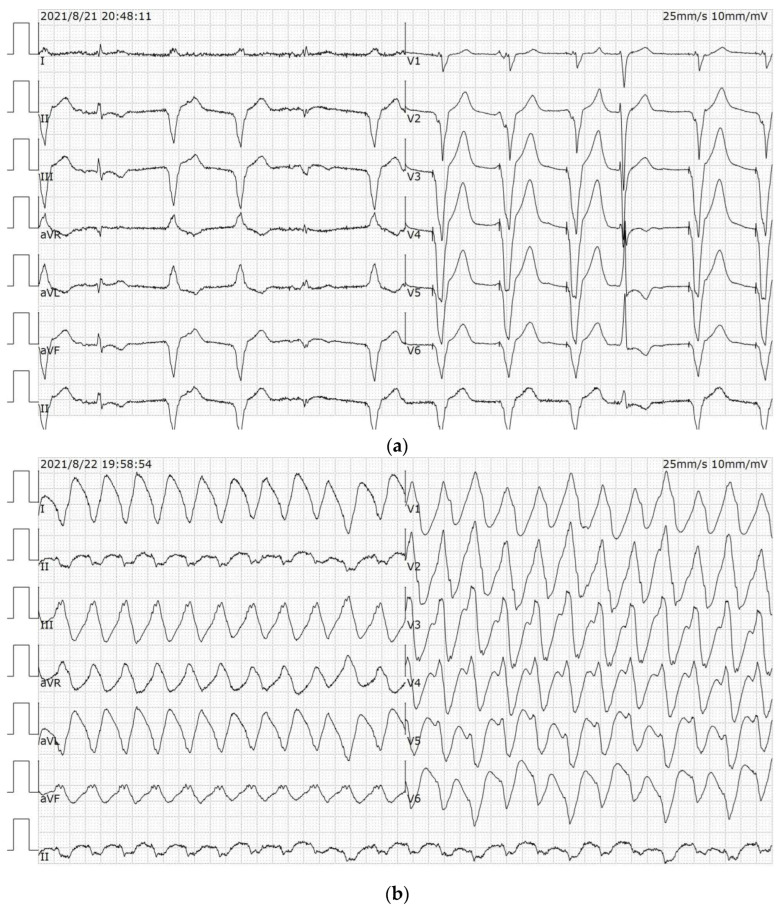
(**a**) Admission ECG shows ventricular pacing rhythm. (**b**) ECG: Ventricular tachycardia (VT) 150 bpm, all of them are fixed circumference, 395 ± 5 ms. (**c**) Sinus rhythm (SR) with atypical intraventricular conduction block and a right-sided electrical axis. (**d**) ECG reviewed after 9 months postoperatively: SR with 55 bpm, normal range.

**Figure 2 diagnostics-12-01955-f002:**
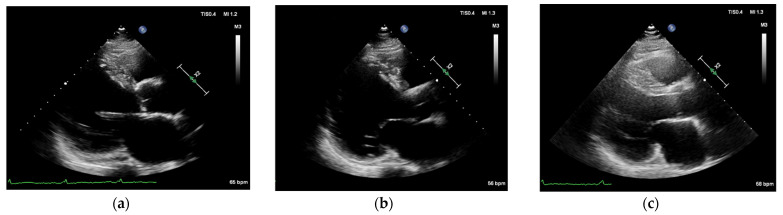
(**a**) VA formation in the posterior left ventricular wall, 4.5 mm wide and 2.3 mm deep; left atrium (55 mm) right atrium (46 mm) left ventricular enlargement (90 mm); hyposystolic left ventricular function; reduced strain on left ventricular wall and uncoordinated activity. Left ventricular end diastolic dimension (LVDd)/left ventricular end-systolic dimension (LVDs): 87/74 mm, EF 29%. (**b**) Left atrium (51 mm) and left ventricle (85 mm) EF 30%. (**c**) VA formation in the posterior left ventricular wall, LVDd/LVDs: 80/66 mm, EF 36%.

**Figure 3 diagnostics-12-01955-f003:**
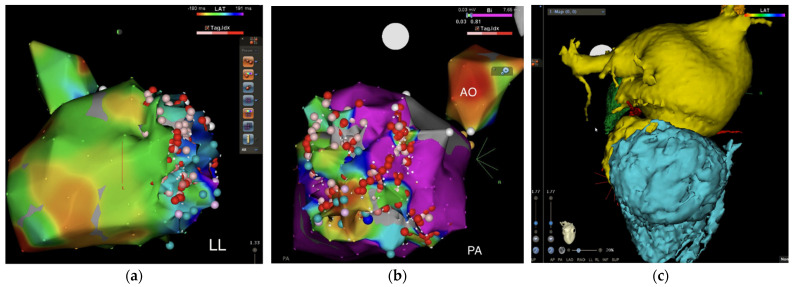
(**a**,**b**) Ablation sites during the procedure: left ventricular cavity hypovoltage zone, late-potential ablation. (**c**) The area of the VA shown after three−dimensional reconstruction of the CCTA.

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
