# Peer review of "Double Gain: The Radio Frequency Catheter Ablation of Ventricular Aneurysm Related Recurrent Ventricular Tachycardia on a Tremendous Cardiac Outpouching"

_diagnostics, 2022, doi:10.3390/diagnostics12081955_

Round 1
Reviewer 1 Report
In this clinical case report the authors describe the treatment
of a patient with extremely enlarged left ventricle, ventricular
aneurysm, LVEF<30%, very high levels of NT-proBNP (>>450 pg/mL),
and recurrent VT episodes (polymorphic) with endocardial
radiofrequency catheter ablation (RFCA) for heart arrhythmias.
After the treatment the LVEF and cardiac function were improved
and the patient was free for VT episodes for the next 9 months.
The authors conclude that RFCA could be used as an effective
diagnostic treatment in this clinical setting.
Concerns
1. This report may contain some valuable information, but more cases
needed to extract broader conclusions.
2. RFCA was used as a therapeutic intervention rather than a diagnostic
treatment.
Author Response
Point 1: This report may contain some valuable information, but more cases
needed to extract broader conclusions.
Response 1:
Thank you for your constructive comment! This case had an giant ventricular aneurysm in the posterior wall of the left ventricle with LVDd of 90 mm. In addition, our case was of VT of left VA origin in a huge heart with endocardial ablation and no recurrence of VT and ICD discharge events at 9 months post-procedure follow-up. This is a case that paradoxes most researches findings and has a successful outcome. Moreover, this case is very rare and has some useful implications for learning.
We have mentioned the limitation of number of cases at the end of the article. In the future, we will keep collecting cases to verify. And studies of endocardial ablation of IDCM combined with VT could be conducted in a joint multicenter setting if it’s possible.
Point 2: RFCA was used as a therapeutic intervention rather than a diagnostic
treatment.
Response 2: Thanks for your constructive point. We have revised our manuscript.
Reviewer 2 Report
Li, Jiang and colleagues in their manuscript report a case of a patient with non-ischemic DCM with recurrent ventricular arrhythmias non-responsive to several antiarrhythmic drugs. The patient underwent successful VT ablation. The field is interesting. The topic is of interest for heart failure and EP specialists. The English style and the discussion should be improved.
I have the following concerns.
Major points
1) The lack of a known genetic tests result should be considered as a limitation.
2) In the present case a ventricular arrhythmia was a severe life threatening manifestation. DCM is known to be very susceptible to any type of arrhythmias. The high risk of arrhythmias, also supraventricular arrhythmias, and their impact even on long term outcomes should be discussed (doi: 10.1016/j.ijcard.2020.08.062).
3) It is not clear if the patient was admitted for cardiac arrest. Please restate.
Minor points
1) The DCM presents several particular findings, including posterior wall aneurism, life threatening arrhythmias and severe LV systolic dysfunction. A speculative hypothesis on the genetic of this patient should be reported.
2) Some discussion on the indication to ICD in patients undergoing VT ablation should be provided (Mayo Clin Proc. 2009 May; 84(5): 483).
3) The manuscript would benefit from a native speaker proofreading
4) Ultrasonic cardiogram is an uncommon wording. Please replace with echocardiography
5) Sotalol administration together with esmolol is quite uncommon. Please discuss this decision.
Author Response
Minor points
- The DCM presents several particular findings, including posterior wall aneurism, life threatening arrhythmias and severe LV systolic dysfunction. A speculative hypothesis on the genetic of this patient should be reported.
Response 1): Thank you very much for this point. We have revised the discussion section related to genetic testing part from line 173. In brief, we have inferred that some of the anti-myocardial antibodies will be positive and the genetic target might be the TNN/LMNA. More details are in the discussion section, please feel free to give us comments.
- Some discussion on the indication to ICD in patients undergoing VT ablation should be provided (Mayo Clin Proc. 2009 May; 84(5): 483).
Response 2): Thank you very much for this constructive comment. We have added some details to the history and changed some misleading descriptions: the patient was first found to have an enlarged heart and HF, and an ICD was implanted in another hospital to prevent sudden cardiac death(Class I, Level of Evidence B)[1]. In the fifth month after discharge from another hospital, VT inevitably developed as the disease progressed and the ICD only prevented the SCD. The patient who had an ICD with optimal anti-arrhythmic and heart failure treatments, and recurrence of the disease still occurred during hospitalization, along with frequent ICD discharges. So,there are indications for RFCA.
- The manuscript would benefit from a native speaker proofreading.
Response 3): Thanks for your point. We have invited a native speaker from Soochow University to help us.
- Ultrasonic cardiogram is an uncommon wording. Please replace with echocardiography.
Response 4): Thanks for your suggestion. We changed it to three-dimensional echocardiography (3D-echo).
- Sotalol administration together with esmolol is quite uncommon. Please discuss this decision.
Response 5): Thanks for your problem. The mechanism for the intermittent intravenous administration of Esmolol in addition to oral Sotalol is that β-blockers reduce sympathetic excitability and play an important role in reducing the recurrence of ventricular tachycardia or ventricular fibrillation[2, 3]. Even if the patient is already on a general β-blocker therapy, an episode of VT or VF requires the addition of an intravenous β-blocker to the original β-blocker in order to block sympathetic hyperexcitability. Therefore, the patient with recurrent VT on the basis of oral Sotalol, intravenous Esmolol can block sympathetic excitation and reduce the recurrence of VT.
In brief, the patient had already been taking Sotalol as a long-term oral β-blocker to improve HF after discharge from another hospital. During her admission in our hospital, she received oral Sotalol and intermittent intravenous infusions of Esmolol, Amiodarone and Lidocaine. After discharge, her oral anti-arrhythmic medication was adjusted to Amiodarone 0.2g qd, metoprolol 47.5mg qd and other drugs to improve heart failure.
( DOI:10.3760/cma.j.cn114798-20201210-01240ï¼›DOI:10.3760/cma.j.cn112138-20201019-00877)
- Pinto YM, Elliott PM, Arbustini E, Adler Y, Anastasakis A, Bohm M, et al. Proposal for a revised definition of dilated cardiomyopathy, hypokinetic non-dilated cardiomyopathy, and its implications for clinical practice: a position statement of the ESC working group on myocardial and pericardial diseases. European heart journal. 2016 Jun 14;37(23):1850-8. PubMed PMID: 26792875.
- The Working Committee on Arrhythmia Drugs CRB, Chinese Society of Biomedical Engineering. Chinese experts suggestions on the application of esmolol injections in patients with arrhythmia. Chinese Journal of Internal Medicine. 2021 (0578-1426 (Print)). chi.
- Association CM, Branch CMACP, Association JotCM, Branch CMAGP, Association ECotCGPJotCM, Institutions EGftPoRDUGfPHC. Guideline for Rational Medication of Ventricular Tachycardia in Primary Care. Chinese Journal of General Practitioners. 2021;20(2):175-83. chi.
Round 2
Reviewer 1 Report
In this clinical case report the authors describe the treatment
of a patient with extremely enlarged left ventricle, ventricular
aneurysm, LVEF<30%, very high levels of NT-proBNP (>>450 pg/mL),
and recurrent VT episodes (polymorphic) with endocardial
radiofrequency catheter ablation (RFCA) for heart arrhythmias.
After the treatment the LVEF and cardiac function were improved
and the patient was free for VT episodes for the next 9 months.
The authors conclude that RFCA could be used as an effective
diagnostic treatment in this clinical setting.
Author Response
We gratefully thank you for your endorsement of our revised manuscipt. And thanks for your previous comments so that we significantly raised the quality of the manuscript and enabled us to improve the manuscript.
Reviewer 2 Report
Thank you for your efforts in addressing my comments. The quality of the manuscript improved.
However, despite an adequate reply in the response to reviewers letter, some points should be better discussed in the revised manuscript.
Major points
2) The importance of ventricular arrhythmias is clearly depicted in the manuscript. However, the role of supraventricular arrhythmias should be discussed and the previous report included amongst references (Int J Cardiol. 2021;323:140-147. doi:10.1016/j.ijcard.2020.08.062)
3) Please clearly report if the patient experienced any symptoms linked to the VT recorded at ICD
Minor points
1) I agree that the most likely genetic mutations are LMNA/FLNC, perhaps in my experience LMNA may be more probable
2) Thank you for the explanation. However, the text should be modified accordingly and the manuscript included amongst references (Mayo Clin Proc. 2009 May; 84(5):483)
Author Response
Major points
2) The importance of ventricular arrhythmias is clearly depicted in the manuscript. However, the role of supraventricular arrhythmias should be discussed and the previous report included amongst references (Int J Cardiol. 2021;323:140-147. doi:10.1016/j.ijcard.2020.08.062)
Response 2) : Thanks for your constructive comment. We have revised manuscript according to your comments in blue from line 174. In brief, when DCM is combined with gene mutations in LMNA, EMD, SCN5A, etc., conduction system disease and supraventricular arrhythmias are susceptible. The presence of supraventricular arrhythmias (premature atrial contraction, atrial tachycardia, atrial flutter (AFL) and atrial fibrillation (AF)), if observed in patients with DCM, should prompt investigation of familial LMNA cardiomyopathy, which is present in 73% of carriers of these genes, compared to 36% of patients without those genetic defects In addition, about 10% of patients with dilated cardiomyopathy present with AF.
3) Please clearly report if the patient experienced any symptoms linked to the VT recorded at ICD
Response 3): Thanks for your point. The part of case reports has been revised in blue. In brief, the patient who was HF and suspected DCM discharged from another hospital with ICD preventing SCD to occur. But in the fifth month after ICD implantation, the patient experienced recurrences of chest tightness and palpitations and was subsequently admitted to our hospital. The patient still experienced recurrences of chest tightness and palpitations, and our team confirmed the recording of the ICD programmer, which showed that there was a series of VT events and the first event was recorded as a frequency of 218 beats per min (bpm) and a duration of 29 seconds. There were 15 repeated ICD discharges with different degrees of distress before the RFCA procedure.
Minor points
- I agree that the most likely genetic mutations are LMNA/FLNC, perhaps in my experience LMNA may be more probable.
Response 1): Thanks for your reply and comment!
2) Thank you for the explanation. However, the text should be modified accordingly and the manuscript included amongst references (Mayo Clin Proc. 2009 May; 84(5):483)
Response 2): Thanks for your point[1]. We have revised manuscript in the references part ([20]).
- Budhraja V. Confirming the diagnosis of cannabinoid hyperemesis. Mayo Clinic proceedings. 2009 May;84(5):483; author reply PubMed PMID: 19411446. Pubmed Central PMCID: PMC2676133. Epub 2009/05/05. eng.
Round 3
Reviewer 2 Report
Thank you for your efforts in addressing my comments. I have no further suggestions.